# Small Differences in Vitamin D Levels between Male Cardiac Patients in Different Stages of Coronary Artery Disease

**DOI:** 10.3390/jcm11030779

**Published:** 2022-01-31

**Authors:** Ewelina A. Dziedzic, William B. Grant, Izabela Sowińska, Marek Dąbrowski, Piotr Jankowski

**Affiliations:** 1Medical Faculty, Lazarski University in Warsaw, 02-662 Warsaw, Poland; 2Sunlight, Nutrition, and Health Research Center, P.O. Box 641603, San Francisco, CA 94164-1603, USA; wbgrant@infionline.net; 3Medical Faculty, Medical University of Warsaw, 02-091 Warsaw, Poland; sowinska.izabela@gmail.com; 4Department of Cardiology, Bielanski Hospital, 01-809 Warsaw, Poland; mardab@wp.pl; 5Department of Internal Medicine and Geriatric Cardiology, Centre of Postgraduate Medical Education, 01-813 Warsaw, Poland; piotrjankowski@interia.pl; 6Institute of Cardiology, Jagiellonian University Medical College, 31-008 Krakow, Poland

**Keywords:** vitamin D, coronary artery disease, myocardial infarction, males, Coronary Artery Surgery Study Score

## Abstract

Cardiovascular diseases are the main cause of mortality in males older than 65 years of age. The prevalent vitamin D deficiency in the worldwide population may have multiple effects on the cardiovascular system. This study sought to determine the association between serum levels of 25-hydroxyvitamin D (25(OH)D) and the stage of coronary artery disease (CAD) in Polish male subjects. Additionally, subjects with a history of myocardial infarction (MI) were analyzed for potential differences in 25(OH)D levels in comparison with those diagnosed with stable CAD. The study was conducted prospectively in a group of 669 male patients subjected to coronarography examination. CAD stage was defined using the Coronary Artery Surgery Study Score. Patients without significant coronary lesions had significantly higher 25(OH)D levels than patients with single-, double-, or triple-vessel disease (median, 17 vs. 15 ng/mL; *p* < 0.01). Significantly lower levels of 25(OH)D were apparent when MI was identified as the cause of the then-current hospitalization in comparison with stable CAD, as well as in patients with a history of MI; all of these cases had lower levels of 25(OH)D in comparison with patients with no such history. Male patients with single-, double-, or triple-vessel CAD, acute coronary syndrome, or a history of MI presented lower serum 25(OH)D.

## 1. Introduction

The aging of the worldwide population is being observed in recent decades. Consequently, the subpopulation of male individuals older than 65 years in Poland has gradually increased [1,2,3,4]. In this group of patients, cardiovascular diseases account for more than half of all deaths [5]. Despite the observed improvement concerning mortality rates, a large difference between the average male and female lifespans persisted (73 and 81 years, respectively) [5]. Elderly men should be considered a unique group of patients liable to major cardiovascular events. Considering exceedingly high rates of the risk of death, cardiovascular diseases (CVDs) continue to prompt the need for the relentless investigation of their risk factors and new therapies [6]. As a result of advancements in treatment, including non-pharmacological prevention, the risk of death from CVDs has dropped by 20% over the last few decades [7]. Despite these improvements, CVD remains responsible for about 18 billion deaths per year, the majority of which are a consequence of myocardial infarction or stroke [8]. Appropriate physical activity and dietary interventions both reduce the cardiovascular risk [9,10] and affect the level of 25-hydroxyvitamin D. Notably, calcitriol deficiency is identified as one of the new cardiovascular disease risk factors and has been proven to be prevalent in human populations worldwide [11,12,13]. 

The discovery of the biological mechanisms underlying vitamin D effects justifies studies of the association between its deficiency and the risk of cardiovascular diseases. The 1,25-dihydroxyvitamin D (calcitriol) receptor is present in numerous cells of the cardiovascular system [14]. Studies conducted on an animal model have shown an adverse effect of vitamin D deficiency on the functions of endothelial cells, vascular smooth muscle, and cardiomyocytes [14]. Due to the presence of the enzyme 1-α hydroxylase, these cells are capable of autocrine calcitriol synthesis [15,16]. This hormone is a negative regulator of the axis of the renin–angiotensin–aldosterone system, the increased activity of which leads to the development of arterial hypertension and myocardial hypertrophy [17,18,19]. The relationship between calcitriol deficiency and individual stages of plaque formation and destabilization, as well as documented risk factors for coronary artery disease, has been documented [13,20]. 

To date, the results of studies do not offer an explicit agreement on how vitamin D affects the cardiovascular system [21]. However, some reports suggest that 25(OH)D levels below the reference range may increase the risk of CVD [21], whereas optimum levels may exert a protective effect on both musculoskeletal and cardiovascular systems [21,22,23]. The main objective of this study was to assess the potential association between serum 25(OH)D levels and the stage of coronary artery disease (CAD) in Polish males. 

## 2. Materials and Methods

### 2.1. Population

This study is part of a research project focused on the relationship between the level of vitamin D and the severity of coronary artery atherosclerosis in Polish cardiac patients. Results of the analysis of this association among 637 patients are presented in previously published articles, in which details of the study population and measurements (diabetes diagnosis, acute coronary syndrome (ACS) diagnosis, interview questionnaire, body mass index (BMI), concentration of total cholesterol (TC) and/or triglycerides (TG), systolic and diastolic blood pressure, coronary angiography, and total 25(OH)D in participant serum and plasma) were described [24,25,26]. Abnormal serum levels of phosphate, calcium and parathyroid hormone (PTH) treatments, or supplementation containing vitamin D or calcium served as exclusion criteria. Patients with stages III-V chronic kidney disease, active malignancy, and elevated inflammatory markers or fever were also excluded from the study. New patients were continuously examined, and final pooled data of Polish patients hospitalized in the Cardiology Department who underwent diagnostic catheter angiography for the evaluation of coronary artery disease in the years 2013 to 2017 were presented in the most previous study [27]. This study presents data for male patients. 

The serum level of 25(OH)D was determined with the Vitamin D total assay by Roche Diagnostics, certified by VDSP. The Roche Diagnostics Vitamin D total assay is a competitive electrochemiluminescence protein-binding assay intended for the quantitative determination of the total 25-OH vitamin D in human serum and plasma. The assay utilizes a vitamin-D-binding protein (VDBP) as the capture protein, which binds to both 25-OH D3 and 25-OH D2 [28].

The status of vitamin D levels was classified according to the Endocrine Society Clinical Practice Guidelines for Vitamin D Deficiency [29]: 25(OH)D level <10 ng/mL was considered as a severe deficiency; ≥10 to <20 ng/mL as a moderate deficiency; ≥20 to <30 ng/mL as a mild deficiency; ≥30 ng/mL as optimal. 

Coronary atherosclerosis was assessed by Coronary Artery Surgery Study Score (CASSS) according to the following rules [30]. Stenosis ≥50% of the left main coronary artery (LMCA) was scored at 2 points. Stenosis >70% in any of the large epicardial coronary arteries (anterior descending branch, LAD; circumflex branch, LCx; right coronary artery, RCA) was scored at 1 point. The sum of all points equaled the score, which may indicate, respectively a one-, two-, or three-vessel CAD [30]. 

The study project was approved by the University Bioethical Committee (KB/124/2014) and followed the rules and principles of the Helsinki Declaration.

### 2.2. Statistics 

The Shapiro–Wilk test was used to evaluate the normal distribution of data. The Poisson regression analysis was used to assess the relationship between 25(OH)D levels and selected variables. To compare the results of continuous variables between the two groups, the Mann–Whitney test or *t*-test was used. For comparisons of three or more independent groups, the Kruskal–Wallis test or one-way analysis of variance were used, depending on the presence of a Gaussian distribution (Shapiro–Wilk normality test). Pearson’s chi-squared test or Fisher’s exact test was used to determine differences between prevalence in selected groups. Statistical analyses were performed with a significance level of 5% (*p* value < 0.05). The statistical analysis was carried out with STATISTICA 12.5 software.

## 3. Results

### 3.1. Characteristics of the Study Group

The study was conducted on 1345 patients admitted to the Department of Cardiology in the Bielanski Hospital in Warsaw (Poland) for coronarographic examination as a result of suspected CAD as identified in an outpatient setting. Final statistical analysis was carried out in 1043 patients (669 males, 374 females). The data of 302 subjects were excluded, as they met the study exclusion criteria (for details, see [27]).

### 3.2. Comparisons between Female and Male Subpopulations

Statistically significant differences were observed between females and males with regard to age, total cholesterol, high-density lipoprotein (HDL) cholesterol, and low-density lipoprotein (LDL) cholesterol levels. Statistically significant disproportions were observed between the female and male subpopulations with regard to smoking status, history of myocardial infarction, and the number of arteries presenting with significant stenosis (Table 1).

Statistically significant differences were observed between females and males with regard to age, total cholesterol, high-density lipoprotein (HDL) cholesterol, and low-density lipoprotein (LDL) cholesterol levels. Statistically significant disproportions were observed between the female and male subpopulations with regard to smoking status, history of myocardial infarction, and the number of arteries presenting with significant stenosis.

### 3.3. Analysis of Male Subpopulation, Correlation between 25(OH)D Levels or Other Parameters, and the CASSS Stage of Coronary Artery Disease in Male Subpopulation

Among male patients divided into individual CASSS severity groups (Table 2), analysis of variance revealed statistically significant differences in age, total cholesterol levels, HDL cholesterol levels, and LDL cholesterol levels.

Post hoc analyses in unequal subgroup populations revealed that CASSS 3 patients were significantly older than patients in all other CASSS groups (*p* < 0.01 for all comparisons). Individual subgroups of patients with different CASSS scores were characterized by statistically significant differences in the prevalence of diabetes, hypertension, smoking, history of myocardial infarction, or the cause of the then ongoing hospitalization.

No significant differences were observed in 25(OH)D levels, BMI values, or triglyceride levels between patients across all CASSS groups. Neither could significant differences be observed in the prevalence of hyperlipidemia or the season of examination between individual CASSS groups. 

Poisson distribution and multiple regression analysis were used to identify factors that significantly determined the CASSS in the male population. Factors/determinants in the analysis included serum 25(OH)D levels, age, BMI, smoking status, hypertension, concomitant diabetes, hyperlipidemia, history of myocardial infarction, cause of the then-current hospitalization, and season during the examination. Factors significantly determining the CASSS value included age (*p* < 0.05), cause of ongoing hospitalization (*p* < 0.001), smoking status (*p* < 0.05), and history of myocardial infarction (*p* < 0.001).

### 3.4. Male Patients without Significant Arterial Stenosis (CASSS 0) in Comparison with Patients with Significant Arterial Stenosis (CASSS 1–3)

The group of patients without significant changes within the coronary arteries (CASSS 0) presented with higher 25(OH)D levels than patients with single-, double-, or triple-vessel disease (CASSS 1, 2, or 3, respectively). Considering the above, we carried out further analyses with the study population divided into two subgroups: the CASSS 0 subgroup and the CASSS 1–3 subgroup (Table 3).

Statistically significant differences were observed between male patients with and without significant coronary stenosis with regard to serum 25(OH)D, BMI values, total cholesterol levels, and HDL cholesterol levels. 

Statistically significant disproportions were observed between subgroups with regard to arterial hypertension status, smoking status, history of myocardial infarction, and cause of the then ongoing hospitalization.

Only 13 patients from the CASSS 0 subgroup were hospitalized for myocardial infarction; the remaining 113 subjects were hospitalized as a result of stable CAD. In addition, 11 patients in the CASSS 0 subgroup had a history of myocardial infarction. About half of the patients from the CASSS 1–3 subgroup were hospitalized for myocardial infarction; about half also had a history of myocardial infarction.

Factors significantly determining the CASSS value in the entire male subpopulation included age (*p* < 0.05), cause of the then-current hospitalization (*p* < 0.001), smoking status (*p* < 0.05), and history of myocardial infarction (*p* < 0.001). Among the CASSS 1–3 subgroup of male patients, larger percentages of patients had arterial hypertension, were smokers, had a history of myocardial infarction, and reported to the department because of an acute coronary syndrome rather than stable CAD.

### 3.5. Determinants of Serum 25(OH)D Levels in Male Cardiac Patients

In the presented study group, significant determinants of serum 25(OH)D levels included the season of the year (*p* < 0.001) and hyperlipidemia (*p* < 0.01). Lower serum 25(OH)D levels were presented independently by patients with hyperlipidemia and those examined between October and April.

### 3.6. Identifying the Group with the Lowest 25(OH)D Levels among Male Cardiac Patients

Factors significantly determining the CASSS value included age (*p* < 0.05), cause of the then-ongoing hospitalization (*p* < 0.001), smoking status (*p* < 0.05), and history of myocardial infarction (*p* < 0.001). The lowest serum 25(OH)D levels were measured in elderly male cardiac patients hospitalized for an acute coronary syndrome, presenting with a history of myocardial infarction, positive smoking status, and diagnosis of hyperlipidemia, as well as undergoing examination between October and April.

## 4. Discussion

Cardiovascular diseases are ranked as the main cause of death in male patients aged over 65 years old, remaining the second in younger individuals [8]. Mortality rates due to CVD for males are higher than those for females [31]. To date, numerous studies suggested a correlation between low 25(OH)D levels and increased risk of death from cardiovascular causes was suggested [32,33]. The aim of our research was to assess the association between vitamin D serum levels the severity of CAD. 

The findings of this research are consistent with the results of previous studies examining the association between serum 25(OH)D levels and CVD [12]. However, only a few studies thus far were conducted in male subpopulations only; most were carried out in mixed-sex populations [34,35,36,37,38]. In our study, male patients without significant coronary lesions (CASSS 0) presented with statistically higher serum vitamin D levels than patients with significant stenosis of coronary arteries (CASSS 1–3); however, the nominal difference was negligible. We showed that calcitriol serum levels were significantly lower in patients with a history of MI. Elderly patients with hyperlipidemia, actively smoking, hospitalized for an ACS, with a history of MI, were a subgroup presenting with the lowest 25(OH)D serum levels. 

The impact of serum vitamin D level on the established cardiovascular risk factors (i.e., development of type 2 diabetes, metabolic syndrome) was repeatedly proven in the literature [39,40]. 

Both nuclear vitamin D receptor (VDR) and the enzyme 25-hydroxyvitamin D3-1α-hydroxylase have been identified in various cells of the cardiovascular system, indicating a direct involvement of this hormone group in the initiation and progression of CVD [41]. Importantly, in patients with heart failure, atrial fibrillation, or coronary artery disease vitamin D deficiency was associated with a worse prognosis [42,43,44]. Moreover, hypovitaminosis D was proven to affect the established cardiovascular risk factors such as arterial hypertension [45], type 2 diabetes [44], or dyslipidemia [44]. Calcitriol inhibits the renin-angiotensin–aldosterone system (RAAS) and the secretion of natriuretic peptides, thus having a hypotensive effect [46]. Activation of the VDR receptor has a protective effect on the excess of angiotensin II by inhibiting fibrosis and exerting anti-inflammatory and antiproliferative effects [47]. Mediated by cells of the immune system, calcitriol modulates the secretion of miR-106b-5p and inhibits the secretion of renin by the glomerular apparatus [48]. The impact of vitamin D on various stages of atherosclerosis is currently being a subject of some studies [13]. Previous studies demonstrated that vitamin D affects atherosclerotic plaque formation in numerous ways, including reducing the inflammatory response, inhibiting the NF-κB pathway [49], and suppressing the post-infarction scar formation [50]. Proper serum 25(OH)D level was proven to reduce the activity of the metalloproteinases, which degrade the fibrous cap of the atherosclerotic plaque [51]. In addition, vitamin D also inhibits the activity of vascular endothelial growth factors, preventing the formation of new vessels within an already formed plaque, thus contributing to its better stability [52]. After plaque rupture, vitamin D exerts an antithrombotic effect by increasing the production of thrombomodulin and reducing the expression of platelet tissue factors. Hence, it inhibits the adhesion of platelets to vascular endothelial cells [53]. This process may be a way of vitamin D contributing to the prevention of ACS.

A study based on an analysis of more than 1000 Polish patients confirmed the already reported low vitamin D levels in the Polish population [54,55], as well as the higher 25(OH)D levels in males than in females [26]. On the other hand, Verdoia emphasizes the importance of higher 25(OH)D levels noted in men, compared with women [56]. The results of that study provided a stimulus to expand the research and reassess how 25(OH)D levels affect the stage of CAD and incidence of MI in the entirely male cohort. The results of the above-mentioned studies stimulated us to evaluate the impact of 25(OH)D levels on the stage of coronary artery disease and the incidence of MI in an all-male cohort. The influence of vitamin D deficiency on episodes of MI in men is supported by several cohort studies [57,58]. Patients with 25(OH)D levels of ≤15 ng/mL were proven to have a more than 1.5-fold increase in the risk of adverse cardiovascular events (i.e., MI, angina pectoris, stroke, TIA, and heart failure) [57] and a twofold increase in the risk of ACS [58]. Moreover, low vitamin D levels and a history of MI were associated with a significant increase in the risk of further major adverse cardiovascular events (MACE), including reoccurrence of MI [37]. In addition, serum 25 (OH) D levels above 7.3 ng/mL were associated with a 40% reduction in the risk of non-fatal MACE in patients with ACS.

To date, several studies examining vitamin D supplementation in patients with CVD have been conducted. Although none of the large cohort studies showed a favorable cardiovascular outcome, individual experiments have proven that six-month calcitriol supplementation significantly reduced the inflammation of coronary arteries [59] and declined SYNTAX score (67). However, poor bioavailability and large intervals between consecutive doses of cholecalciferol should be underlined as potentially resolvable issues [59]. Another possible mechanism by which vitamin D may affect the degree of progression of coronary disease is its effect on the metabolism of sex hormones. In the MESA study, lower 25(OH)D levels were found to be associated with lower sex-hormone-binding globulin concentrations and higher levels of free testosterone, which are important in the course of coronary artery disease [60].

Our research suffers from several limitations. The study group consisted of residents of only central Poland, most residing in urban areas. Expanding the study group to include residents of other provinces would facilitate the translation of the results to the entire Polish population. CAD staging was classified based on the results of coronary angiography using the CASSS. The classification of the severity of atherosclerosis based on the SYNTAX might change our results.

The results of the observational studies carried out so far have shown that the endocrine system of vitamin D, in addition to its documented effect on the skeletal system, exerts a wide spectrum of extra-skeletal activity [61,62]. In these studies, low vitamin D levels were found to be associated with an increased risk of cardiovascular diseases, including hypertension, congestive heart failure, as well as adverse cardiovascular events (MACE, heart attacks, and strokes). In a meta-analysis involving nearly 850,000 people, low serum 25(OH)D levels were associated with a 1.42 times higher risk of developing MACE, compared with patients with higher levels of vitamin D [63]. On the other hand, the results of randomized clinical trials (VITAL, ViDa, D2d), which included over 30,000 participants, showed that supplementation with vitamin D does not prevent cardiovascular events or the progression of type 2 diabetes [64,65,66]. It should be emphasized that the initial serum level of 25 (OH) D in the respondents of the above-mentioned studies fluctuated above 50 nmol/L, and post hoc analysis suggested some extra-skeletal benefits in the vitamin D deficiency group. The causal association between calcitriol and cardiovascular mortality continues to be the subject of much debate. New information was provided by the recently presented results of the non-linear MR analysis carried out at UK Biobank [67]. The authors of the cited study presented the association between the genetically predicted serum 25 (OH) D levels and the risk of cardiovascular diseases to be L shaped. This research seems to confirm the results of observational and interventional studies and determines a specific range of vitamin D levels within which vitamin D supplementation may have a beneficial effect in short- and long-term observations. At the same time, it explains why supplementing people rich in vitamin D does not generate overall health benefits, and correction of a severe deficiency of this hormone may be necessary. At present, the opinions of scientists around the world unanimously recommend the correction of vitamin D (25 (OH) D serum deficiency <30 nmol/L), and most scientific societies recommend a target level of >50 nmol/L as optimal for bone health. In our opinion, vitamin D deficiency may also be an easily modifiable risk factor of the acute coronary syndrome in men, which should undoubtedly be the subject of further research. Perhaps, well-designed and conducted social campaigns in the field of proper exposure to solar radiation, food fortification, or pharmacological supplementation of vitamin D could considerably contribute to the prevention of CAD and its complications.

## 5. Conclusions

In conclusion, we demonstrated that male patients with a history of ACS and MI presented reduced serum calcitriol levels. Patients with advanced CAD presented with significantly lower levels of 25(OH)D than those without significant atherosclerotic lesions; however, the difference should be considered as clinically negligible. Further studies should be undertaken in specific subgroups, to assess the potential beneficial effects of vitamin D supplementation in this group of patients.

## Figures and Tables

**Table 1 jcm-11-00779-t001:** Results between female and male subpopulations.

Variable	Females	Males	*p*
*N*	374	669	N/A
Age (years)	70 ± 11	65 ± 11	<0.001
Body mass index (kg/m^2^)	28 ± 5	28 ± 5	0.75
Body mass index class (1/2/3) ^†^	100/120/131	156/274/197	<0.05
Diabetes (No/Yes/Prediabetes)	242/120/12	418/225/26	0.72
Total cholesterol (mg/Dl)	189± 49	172 ± 46	<0.001
High-density lipoprotein (mg/dL)	55 ± 16	46 ± 14	<0.001
Low-density lipoprotein (mg/dL)	108 ± 44	100 ± 41	<0.01
Triglycerides (mg/dL)	129± 57	130 ± 74	0.82
Hyperlipidemia (No/Yes)	131/218	265/348	0.08
Hypertension (No/Yes)	57/317	124/545	0.18
Smoking (No/Yes/Ex)	272/72/30	371/221/77	<0.001
History of myocardial infarction (No/Yes)	254/120	389/280	<0.01
Cause of hospitalization (0/1) ^‡^	232/141	389/277	0.23
Coronary Artery Surgery Study Score (0/1/2/3)	142/96/74/62	126/180/208/155	<0.001
Level of 25-hydroxyvitamin D (ng/mL (range))	14 (4–55)	16 (4–48)	0.07
25(OH)D level (1/2/3/4) **	101/168/76/29	128/357/149/35	<0.01
Season (October–April/May–September)	291/83	494/175	0.16

25(OH)D = 25-hydroxyvitamin D; BMI, body mass index; CASSS, Coronary Artery Surgery Study Score; HDL, high-density lipoprotein; LDL, low-density lipoprotein. ^†^ BMI class 1, <25; class 2, 25–30; class 3, >30; ^‡^ 0, stable coronary artery disease; 1, myocardial infarction; ** 1: <10 ng/mL severe deficiency, 2: ≥10 to <20 ng/mL moderate deficiency, 3: ≥20 to <30 ng/mL mild deficiency, 4: ≥30 ng/mL optimal.

**Table 2 jcm-11-00779-t002:** Characteristics of the examined group divided according to degree of coronary atherosclerosis (the Coronary Artery Surgery Study Score (CASSS)) into four subgroups.

Variable	CASSS 0	CASSS 1	CASSS 2	CASSS 3	*p*
*N*	126	180	208	155	
Age (years)	64 ± 11	64 ± 12	64 ± 10	68 ± 10	<0.001
Body mass index (kg/m^2^)	29 ± 6	28 ± 5	28 ± 5	28 ± 5	0.07
Body mass index class (1/2/3) ^†^	30/45/47	38/71/53	49/93/55	39/65/42	0.50
Diabetes (No/Yes/Prediabetes)	88/34/4	122/52/6	123/73/12	85/66/4	<0.05
Total cholesterol (mg/dL)	180 ± 45	176 ± 44	171 ± 49	163 ± 45	<0.05
High-density lipoprotein (mg/dL)	52 ± 18	46 ± 13	46 ± 13	43 ± 12	<0.001
Low-density lipoprotein (mg/dL)	104 ± 38	105 ± 40	100 ± 43	92 ± 39	<0.05
Triglycerides (mg/dL)	124 ± 68	127 ± 66	130 ± 68	137 ± 93	0.54
Hyperlipidemia (No/Yes)	43/66	61/107	90/103	71/71	0.07
Hypertension (No/Yes)	34/92	36/144	29/179	25/130	<0.05
Smoking (No/Yes/Ex)	88/26/12	90/74/16	104/72/32	89/49/17	<0.01
History of myocardial infarction (No/Yes)	115/11	112/68	96/112	66/89	<0.001
Cause of hospitalization (0/1) ^‡^	113/13	86/93	115/91	75/80	<0.001
Level of 25-hydroksyvitamin D (ng/mL (range))	17 (5–47)	15 (4–48)	15 (4–37)	15 (4–43)	0.05
25(OH)D level (1/2/3/4) **	19/64/35/8	36/102/32/10	35/124/41/8	38/67/41/9	0.08
Season (October–April/May–September)	96/30	129/51	157/51	112/43	0.73

25(OH)D = 25-hydroxyvitamin D; BMI, body mass index; CASSS, Coronary Artery Surgery Study Score; HDL, high-density lipoprotein; LDL, low-density lipoprotein. ^†^ BMI class 1: <25; class 2: 25–30; class: 3 >30; ^‡^ 0, stable coronary artery disease; 1, myocardial infarction; ** 1: <10 ng/mL severe deficiency, 2: ≥10 to <20 ng/mL moderate deficiency, 3: ≥20 to <30 ng/mL mild deficiency, 4: ≥30 ng/mL optimal.

**Table 3 jcm-11-00779-t003:** Characteristics of the examined group divided according to degree of coronary atherosclerosis (CASSS) into two subgroups.

Variable	CASSS 0	CASSS 1–3	*p*
*N*	126	543	
Age (years]	64 ± 11	65 ± 11	0.78
Body mass index (kg/m^2^)	29 ± 5	28 ± 5	<0.01
Body mass index class (1/2/3) ^†^	30/45/47	126/229/150	0.13
Diabetes (No/Yes/Prediabetes)	88/34/4	330/191/22	0.17
Total cholesterol (mg/dL)	180 ± 45	170 ± 46	<0.05
High-density lipoprotein (mg/dL)	52 ± 18	45 ± 13	<0.001
Low-density lipoprotein (mg/dL)	104 ± 38	99 ± 41	0.28
Triglycerides (mg/dL)	124 ± 68	131 ± 75	0.38
Hyperlipidemia (No/Yes)	43/66	222/282	0.38
Hypertension (No/Yes)	34/92	90/453	<0.01
Smoking (No/Yes/Ex)	88/26/12	283/195/65	<0.01
History of myocardial infarction (No/Yes)	115/11	274/269	<0.001
Cause of hospitalization (0/1) ^‡^	113/13	276/264	<0.001
Level of 25-hydrokyvitamin D (ng/mL (range))	17 (5–47)	15 (4–48)	<0.01
25(OH)D level (1/2/3/4) **	19/64/35/8	109/293/114/27	0.26
Season (October–April/May–September)	96/30	398/145	0.51

^†^ BMI class 1 < 25; class 2 25–30; class 3 > 30; ^‡^ 0, stable coronary artery disease; 1, myocardial infarction; ** 1: <10 ng/mL severe deficiency, 2: ≥10 to <20 ng/mL moderate deficiency, 3: ≥20 to <30 ng/mL mild deficiency, 4: ≥30 ng/mL optimal.

## Data Availability

Data can be provided by the authors upon reasonable request.

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
