# Peer review of "Small Differences in Vitamin D Levels between Male Cardiac Patients in Different Stages of Coronary Artery Disease"

_jcm, 2022, doi:10.3390/jcm11030779_

Round 1

Reviewer 1 Report

There are several important limitations of this work.

  1. Patients with coronary artery disease (CAD) have lower levels of 25-hydroxyvitamin D than patients without the coronary disease (15 ng/ml (4-48 ng/ml) vs 17 ng/ml (5-48 ng/ml).

However, the difference is extremely small with a complete overlap between values of vitamin 25(OH)D. Such a small difference which is less than the coefficient of variation for the laboratory measurement of vitamin 25(OH)D in the blood is not relevant.

  1. The second main result is that patients with MI and/or with a history of MI (I don’t understand from the text and see these results in the Results section) have lower levels of 25(OH)D. If the authors enrolled patients with ACS it is not appropriate since these patients have completely different metabolism influenced by the inflammatory and hemodynamic changes during acute disease.  
  2. The use of simple stratification of no, one-, two- and three-vessel disease is not appropriate for the extent of coronary atherosclerotic disease. SYNTAX score is now a standard for the estimation of the extent of CAD. The correlation between 25(OH)D and Syntax score or even Syntax II score would be more relevant for the association between 25(OH)D levels and coronary disease.
  3. Since the large meta-analysis included more than 83 000 participants, has shown that vitamin D supplementation was not associated with reduced risks of major adverse cardiovascular events, myocardial infarction, stroke, cardiovascular disease mortality, or all-cause mortality compared with placebo, it is important to make a proper investigation and to examine the potential role of vitamin D for CAD and to investigate different confounding factors.
  4. This MS is not close to being published in JCM. Major, Major revision should be provided.

Author Response

Thank you for giving us the opportunity to submit a revised draft of the manuscript "Vitamin D May Affect the Stage of Coronary Artery Disease and Myocardial Infarction Episodes in Male Patients". A comprehensive summary of our manuscript and insightful comments let us introduce valuable corrections. Below please find a point-by-point response to reviewer and editor comments.

  1. Patients with coronary artery disease (CAD) have lower levels of 25-hydroxyvitamin D than patients without the coronary disease (15 ng/ml (4-48 ng/ml) vs 17 ng/ml (5-48 ng/ml). However, the difference is extremely small with a complete overlap between values of vitamin 25(OH)D.

We thank the Reviewer for underlining this important issue. The only patients qualified for the study were those with CAD confirmed with angiography. The serum concentration of vitamin D difference was statistically significant between patients without any significant changes in coronary arteries and those with severe stenosis of at least one main coronary artery (CASSS 0 - 17 ng/ml vs CASSS 1-3 15 ng/ml). We agree with the Reviewer that the significance of our finding that vitamin D level correlate significantly with the extant of coronary atherosclerosis is unclear. Therefore, we emphasized in the main text that the difference was clinically negligible. Moreover, we decided to cross out from the conclusions our claim that vitamin D should be considered as a potential modifiable risk factor for ACS and coronary artery disease. On the other hand, our results should be seen as a step in the research concerning factors related to atherosclerosis development and cardiovascular risk. Consequently, our results may convince other scholars to investigate the relation between vitamin D and atherosclerosis.

  1. Such a small difference which is less than the coefficient of variation for the laboratory measurement of vitamin 25(OH)D in the blood is not relevant.

We thank the Reviewer for this comment. It is widely accepted in the international scientific community that 25(OH)D serum concentration is the best indicator of its reserve in patients [1,2]. It was concluded that mass spectrometry (HPLC/MSMS) or immunometric methods are viable for clinical quantification of vitamin D [2]. Vitamin D Standardization Program (VDSP), operating since 2010, publishes a list of certified manufacturers yearly. The analytic methods developed by those manufacturers are deemed to be compliant to the standards [3]. In our study we used Vitamin D total test by Roche Diagnostics, certified by VDSP. The Roche Diagnostics Vitamin D total assay is a competitive electrochemiluminescence protein binding assay intended for the quantitative determination of total 25-OH vitamin D in human serum and plasma. The assay utilizes a vitamin D binding protein (VDBP) as the capture protein, which binds to both 25-OH D3 and 25-OH D2 (Roche Diagnostics, Mannheim, Germany) [4]. In 2021 Trimboli et al. confirmed the reliability of Elecsys Vitamin D Total II test (Roche Diagnostics, Mannheim, Germany) in routine diagnostic utility setting with coefficient of variation CV below 5% [5]. This was also proven previously in 2013 by Abdel-Wareth et al. [6]. The difference shown by our data is bigger than said 5%, thus bigger than the coefficient of variation. The authors hope the Reviewer will accept the explanation.

[1] M.F. Holick, N.C. Binkley, H.A. Bischoff-Ferrari, C.M. Gordon, D.A. Hanley, R.P. Heaney, M.H. Murad, C.M. Weaver, Evaluation, treatment, and prevention of vitamin D deficiency: an endocrine society clinical practice guideline, J.Clin. Endocrinol. Metabol. 96 (2011) 1911–1930, https://doi.org/10.1210/jc.2011-0385.

[2] N. Heureux, Vitamin D Testing—Where Are We and what Is on the Horizon?, 2017, pp. 59–101, https://doi.org/10.1016/bs.acc.2016.07.002

[3] Laboratory Quality Assurance and Standardization Programs, VDSCP List of Certified Participants | CDC, (n.d.). https://www.cdc.gov/labstandards/vdscp.html (accessed on January 29, 2021).

[4] Cobas E411 Vitamin D Total Reagent Insert (07464215190V2.0), Roche Diagnostics. [(accessed on 19 January 2022)]. Available online: http://labogids.sintmaria.be/sites/default/files/files/vit._d_total_ii_2017-11_v2.pdf

[5] Francesca Trimboli, Salvatore Rotundo, Simone Armili, Selena Mimmi, Fortunata Lucia, Nicola Montenegro, Giulio Cesare Antico, Alessandro Cerra, Maria Gaetano, Francesco Galato, Lorenza Giaquinto Carinci, Danilo Iania, Serafina Mancuso, Maria Martucci, Consuelo Teti, Marta Greco, Giovanni Cuda, Elvira Angotti, Serum 25-hydroxyvitamin D measurement: Comparative evaluation of three automated immunoassays, Practical Laboratory Medicine,Volume 26,2021,e00251 ISSN 2352-5517,https://doi.org/10.1016/j.plabm.2021.e00251.

[6] Abdel-Wareth L, Haq A, Turner A, et al. Total vitamin D assay comparison of the Roche Diagnostics Vitamin D total electrochemiluminescence protein binding assay with the Chromsystems HPLC method in a population with both D2 and D3 forms of vitamin D. Nutrients. 2013;5(3):971-980. Published 2013 Mar 22. doi:10.3390/nu5030971

3. The second main result is that patients with MI and/or with a history of MI (I don’t understand from the text and see these results in the Results section) have lower levels of 25(OH)D. If the authors enrolled patients with ACS it is not appropriate since these patients have completely different metabolism influenced by the inflammatory and hemodynamic changes during acute disease.

We thank the Reviewer for underlining this important issue. To clarify, patients hospitalized with acute coronary syndrome as a primary reason had significantly lower vitamin D concentration in comparison to patients hospitalized with coronary artery disease as a primary reason. Similarly, patients with myocardial infarction in the history had significantly lower vitamin D concentration than those without history of MI.

The core of our study is based on the influence of vitamin D on the formation and destabilization of atherosclerotic plaque as well as the link between said hormone and CAD risk factors, both previously stated by plethora of authors.

We agree that the continuum of atherosclerosis causing CAD, resulting with MI in some patients, is a complicated, multi-stage inflammatory process of blood vessel wall [1]. However, in our opinion, the vitamin D serum concentration does not, in fact, depend on clinical characteristic of patients, which are undeniably distinct for patients with CAD or ACS. We specifically addressed the factors that influence the vitamin D levels at the planning stage of our study. We excluded patients with chronic kidney disease (stage III-V) due to concomitant disturbances of calcium‐phosphate metabolism, cancer diagnosis (paraneoplastic syndromes and associated disorders of calcium-phosphate metabolism), elevated markers of inflammation (total white blood cell count >10,000 cells/μl or C-reactive protein concentration > 5 mg/l) or fever, and treated with medications or supplements containing vitamin D or calcium. For each patient, serum phosphate, calcium, and parathyroid hormone concentrations were measured. Participants with abnormal results of aforementioned tests were excluded. We did also verify that all patients were treated with comparable doses of either atorvastatin or rosuvastatin .

Exclusion criteria were added to Materials and methods.

[1] Christodoulidis, Georgios; Vittorio, Timothy J.; Fudim, Marat; Lerakis, Stamatios; Kosmas, Constantine E. Inflammation in Coronary Artery Disease, Cardiology in Review: November/December 2014 - Volume 22 - Issue 6 - p 279-288.

4. The use of simple stratification of no, one-, two- and three-vessel disease is not appropriate for the extent of coronary atherosclerotic disease. SYNTAX score is now a standard for the estimation of the extent of CAD. The correlation between 25(OH)D and Syntax score or even Syntax II score would be more relevant for the association between 25(OH)D levels and coronary disease.

We thank the Reviewer for underlining this important issue. We agree that SYNTAX score is now the standard for the estimation of the extent of CAD. These arguments contributed to our choice to utilize CASS scale. Moreover, our choice was based on the study by Neeland et al., who compared 10 most popular scales, namely Gensini, CASS, Duke CAD Severity Index, Syntax, Duke Jeopardy, BARI Jeopardy Index 100, Jenkins, Friesinger, Sullivan, and Approach in assessment of 3600 coronary catheterizations [1]. Furthermore, in a subset of 50 patients, they quantified plaque burden and plaque area in the left anterior descending coronary artery using intravascular ultrasound (IVUS). Their conclusion is that all compared angiographic scores correlated with each other and all scores correlated significantly with average plaque burden and plaque area by IVUS. The CASS-50 score had the strongest correlation and correlated with angiographic scores. The recent study by Kim et al. compared the innovative approach of fractional myocardial mass (FMM) - a vessel-specific myocardial mass - with angiographic scores (APPROACH, Duke Jeopardy, BARI, CASS, SYNTAX, Jenkins, BCIS-1, Leaman, Modified Duke) and investigated its discriminative performance for the presence of myocardial ischemia defined by invasively measured fractional flow reserve (FFR) for predicting myocardial ischemia [2]. The FMM was comparable to angiographic scores for prediction of functional stenosis defined by FFR≤0.80. This result further supports our choice of CASS over SYNTAX, as these systems seem to be of comparable value for the CAD severity assessment in coronary catheterizations. Indeed, the advantage of the SYNTAX score over the CASS scale can be seen in the clinical setting, when the plaque localization, morphology, the presence of bifurcations, etc. is important, however, the advantage disappears to a large extent when used to estimate the extant o coronary atherosclerosis. Nevertheless, we are fully aware that the utilization of SYNTAX or SYNTAX II score would be more beneficial in the scientific context, hence we are pointing it out in Limitations.

[1] Neeland IJ, Patel RS, Eshtehardi P, Dhawan S, McDaniel MC, Rab ST, Vaccarino V, Zafari AM, Samady H, Quyyumi AA. Coronary angiographic scoring systems: an evaluation of their equivalence and validity. Am Heart J. 2012 Oct;164(4):547-552.e1. doi: 10.1016/j.ahj.2012.07.007. PMID: 23067913; PMCID: PMC3913177.

[2] Kim HY, Doh JH, Lim HS, Nam CW, Shin ES, Koo BK, Lee JM, Park TK, Yang JH, Song, YB, Hahn JY, Choi SH, Gwon HC, Lee SH, Kim SM, Choe Y, Choi JH. Comparison of fractional myocardial mass, a vessel-specific myocardial mass-at-risk, with coronary angiographic scoring systems for predicting myocardial ischemia. J Cardiovasc Comput Tomogr. 2020 Jul-Aug;14(4):322-329. doi: 10.1016/j.jcct.2019.11.001. Epub 2019 Nov 25. PMID: 31786052.

5. Since the large meta-analysis included more than 83000 participants, has shown that vitamin D supplementation was not associated with reduced risks of major adverse All scores correlated significantly with average plaque burden and plaque area by IVUS (range ρ: 0.56-0.78, p<0.0001 and 0.43-0.62, p<0.01, respectively cardiovascular events, myocardial infarction, stroke, cardiovascular disease mortality, or all-cause mortality compared with placebo, it is important to make a proper investigation and to examine the potential role of vitamin D for CAD and to investigate different confounding factors.

The authors are grateful the Reviewer for this great comment. The results of the observational studies carried out so far have shown that the endocrine system of vitamin D, in addition to its documented effect on the skeletal system, exerts a wide spectrum of extra-skeletal activity. In these studies, low vitamin D levels were found to be associated with an increased risk of cardiovascular diseases, including hypertension, congestive heart failure as well as adverse cardiovascular events (MACE, heart attacks and strokes). In a meta-analysis involving nearly 850,000 people, low serum 25 (OH) D levels were associated with a 1.42 times higher risk of developing MACE compared to patients with higher levels of vitamin D. On the other hand, the results of randomized clinical trials (VITAL, ViDa, D2d), which included over 30,000 participants, showed that supplementation with vitamin D does not prevent cardiovascular events or the progression of type 2 diabetes. It should be emphasized that the initial serum concentration of 25 (OH) D in the respondents of the above-mentioned studies fluctuated above 50 nmol/l, and post hoc analysis suggested some extra-skeletal benefits in the vitamin D deficiency group. The causal association between calcitriol and cardiovascular mortality continues to be the subject of much debate. New information was provided by the recently presented results of the non-linear MR analysis carried out at UK Biobank. The authors of the citied study present the association between the genetically predicted serum 25 (OH) D concentration and the risk of cardiovascular diseases to be L-shaped. This work seems to confirm the results of observational and interventional studies and determines a specific range of vitamin D concentration within which vitamin D supplementation may have a beneficial effect in short and long-term observation. At the same time, it explains why supplementing people rich in vitamin D does not generate overall health benefits; and correction of a severe deficiency of this hormone may be necessary. At present, the opinions of scientists around the world unanimously recommend the correction of vitamin D (25 (OH) D serum deficiency <30 nmol/l), and most scientific societies recommend a target concentration of > 50 nmol/l as optimal for bone health. In our opinion, vitamin D deficiency may also be an easily modifiable risk factor of acute coronary syndrome in men, which should undoubtedly be the subject of further research. Perhaps, well-designed and conducted social campaigns in the field of proper exposure to solar radiation, food fortification or pharmacological supplementation of vitamin D could play a large role in the prevention of CAD and its complications. The authors fully agree with the Reviewer we need more well designed research in this area to examine the potential role of vitamin D for CAD and to investigate different confounding factors. Indeed, our results should be seen as a potential premise for further research.

6. This MS is not close to being published in JCM. Major, Major revision should be provided.

The authors thank the Reviewer very much for assessment of our manuscript and for suggestions improving the paper. We sincerely hope that our explanations will convince the Reviewer.

Reviewer 2 Report

Interesting  paper  on a relevant  topic.

The  role  of vitamin D in  coronary  artery  disease  is  well analyzed and  described.

In the  methods  section  the Gensini score  should be  described in detail (even if it is cited in the  references).

In  the  results  section   the  cut-off value of normal Vitamin D , as threshold for  calcitriol  administration,  should  be  used  to assess how  many  patients  are  below  or  above    this  value  and  commented  in the  discussion section

Author Response

Thank you for giving us the opportunity to submit a revised draft of the manuscript "Vitamin D May Affect the Stage of Coronary Artery Disease and Myocardial Infarction Episodes in Male Patients". A comprehensive summary of our manuscript and insightful comments let us introduce valuable corrections. Below please find a point-by-point response to reviewer and editor comments.

  1. In the methods section the Gensini score should be described in detail (even if it is cited in the references).

Answer: According to the Reviewer's suggestion, we added a description of the CASS score to the methodology section.

  1. In the results section the cut-off value of normal Vitamin D , as threshold for calcitriol administration, should be used to assess how many patients are below or above this value and commented in the discussion section.

Answer: According to the Reviewer's suggestion, in the results section, we presented the division of the analyzed group of patients using the classification of the degree of vitamin D deficiency recommended by the European Society of Endocrinology.